# SeqXGPT: Sentence-Level AI-Generated Text Detection

**Pengyu Wang, Linyang Li** [*] **, Ke Ren, Botian Jiang,**
**Dong Zhang, Xipeng Qiu** [†]
School of Computer Science, Fudan University
Shanghai Key Laboratory of Intelligent Information Processing, Fudan University
{pywang22,kren22,btjiang23,dongzhang22}@m.fudan.edu.cn
{linyangli19,xpqiu}@fudan.edu.cn

## Abstract

Widely applied large language models (LLMs) can generate human-like content, raising concerns about the abuse of LLMs. Therefore, it is important to build strong AI-generated text (AIGT) detectors. Current works only consider document-level AIGT detection, therefore, in this paper, we first introduce a sentence-level detection challenge by synthesizing a dataset that contains documents that are polished with LLMs, that is, the documents contain sentences written by humans and sentences modified by LLMs. Then we propose **Seq**uence **X** (Check) **GPT**, a novel method that utilizes log probability lists from white-box LLMs as features for sentence-level AIGT detection. These features are composed like *waves* in speech processing and cannot be studied by LLMs. Therefore, we build SeqXGPT based on convolution and self-attention networks. We test it in both sentence and document-level detection challenges. Experimental results show that previous methods struggle in solving sentence-level AIGT detection, while our method not only significantly surpasses baseline methods in both sentence and document-level detection challenges but also exhibits strong generalization capabilities.[1]

## 1 Introduction

With the rapid growth of Large Language Models (LLMs) exemplified by PaLM, ChatGPT and GPT-4 (Chowdhery et al., 2022; OpenAI, 2022, 2023), which are derived from pre-trained models (PTMs) (Devlin et al., 2018; Radford et al., 2019; Lewis et al., 2019; Raffel et al., 2020), AI-generated texts (AIGT) can be seen almost everywhere. Therefore, avoiding the abuse of AIGT is a significant challenge that require great effort from NLP researchers. One effective

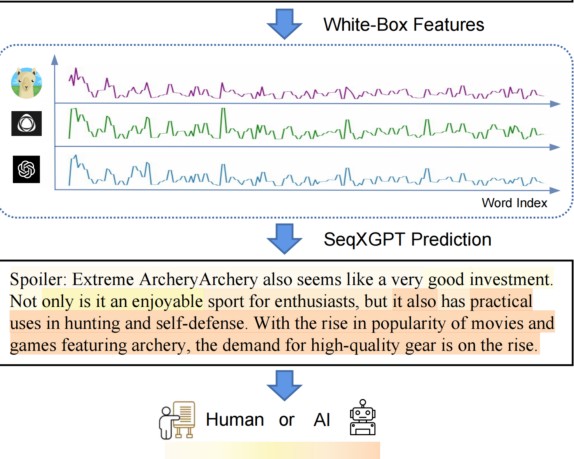

Figure 1: Process of Sentence-Level AIGT Detection Challenge. In this figure, only the **first sentence** of the candidate document is **human-created**, while the others are generated by AI (GPT-3.5-turbo). The deeper the color, the higher the likelihood that this part is AI-generated. The example demonstrates that SeqXGPT can efficiently conduct sentence-level AIGT detection.

solution is AIGT detection, which discriminates whether a given text is written or modified by an AI system (Mitchell et al., 2023; Li et al., 2023).

Current AIGT detection strategies, such as supervised-learned discriminator [2], perplexity-based methods (Mitchell et al., 2023; Li et al., 2023), etc., focus on discriminating whether a whole document is generated by an AI. However, users often modify partial texts with LLMs rather than put full trust in LLMs to generate a *whole document*. Therefore, it is important to explore fine-grained (e.g. sentence-level) AIGT detection.

Building methods that solve the sentence-level AIGT detection challenge is **not** an incremental modification over document-level AIGT detection. On the one hand, model-wise methods like DetectGPT and Sniffer require a rather long

---

[*]Advisor.
[†]Corresponding author.
[1]Our code and generated datasets are available at https://github.com/Jihuai-wpy/SeqXGPT

[2]https://github.com/openai/gpt-2-output-dataset

document as input (over 100 tokens), making them less effective at recognizing short sentences. On the other hand, supervised methods such as RoBERTa fine-tuning, are prone to overfitting on the training data, as highlighted by Mitchell et al. (2023); Li et al. (2023). Therefore, it is necessary to build sentence-level detectors considering the limitation of current AIGT methods.

In this paper, we study the sentence-level AIGT detection challenge:

We first build a sentence-level AIGT detection dataset that contains documents with both human-written sentences and AI-generated sentences, which are more likely to occur in real-world AI-assisted writings. Further, we introduce **SeqXGPT**, a strong approach to solve the proposed challenge. Since previous works (Solaiman et al., 2019a; Mitchell et al., 2023; Li et al., 2023) show that perplexity is a significant feature to be used for AIGT detection, we extract word-wise log probability lists from white-box models as our foundational features, as seen in Figure 1. These temporal features are composed like *waves* in speech processing where convolution networks are often used (Baevski et al., 2020; Zhang et al., 2023). Therefore, we also use convolution network, followed by self-attention (Vaswani et al., 2017) layers, to process the wave-like features. We employ a sequence labeling approach to train SeqXGPT, and select the most frequent word-wise category as the sentence category. In this way, we are able to analyze the fine-grained sentence-level text provenance.

We conduct experiments based on the synthesized dataset and test modified existing methods as well as our proposed SeqXGPT accordingly. Experimental results show that existing methods, such as DetectGPT and Sniffer, fail to solve sentence-level AIGT detection. Our proposed **SeqXGPT** not only obtains promising results in both sentence and document-level AIGT detection challenges, but also exhibits excellent generalization on out-of-distribution datasets.

## 2 Background

The proliferation of Large Language Models (LLMs) such as GPT-4 has given rise to increased concerns about the potential misuse of AI-generated texts (AIGT) (Brown et al., 2020; Zhang et al., 2022; Scao et al., 2022). This emphasizes the necessity for robust detection mechanisms to ensure security and trustworthiness of LLM applications.

### 2.1 Principal Approaches to AIGT Detection

Broadly, AIGT detection methodologies fall into two primary categories:

1. **Supervised Learning:** This involves training models on labeled datasets to distinguish between human and AI-generated texts. Notable efforts in this domain include the studies by Bakhtin et al. (2019); Solaiman et al. (2019b); Uchendu et al. (2020), where supervised models were trained on the top of neural representations. OpenAI releases GPT-2 outputs along with a supervised-learning baseline based on RoBERTa, which claims that GPT-2 outputs are complicated for pre-trained models to discriminate.

2. **Utilizing Model-wise Features:** Rather than relying solely on labeled datasets, this approach focuses on intrinsic features of texts, such as token log probabilities, token ranks, and predictive entropy, as proposed by Gehrmann et al. (2019); Solaiman et al. (2019b); Ippolito et al. (2020). The original version of GPTZero [3] attempts to use the text perplexities as white-box features, while DetectGPT (Mitchell et al., 2023) innovated a perturbation-based method leveraging average per-token log probabilities. The recent work of Li et al. (2023) extends this further by studying multi-model detection through text perplexities, aiming to trace the precise LLM origin of texts.

### 2.2 Detection Task Formulations

The AIGT detection tasks can be formulated in different setups:

1. **Particular-Model Binary AIGT Detection:** In this setup, the objective is to discriminate whether a text was produced by a *specific* known AI model or by a human. Both GPTZero and DetectGPT fall into this category.

2. **Mixed-Model Binary AIGT Detection:** Here, the detectors are designed to identify AI-generated content without the need to pinpoint the exact model of origin.

---

[3] https://gptzero.me/

3. **Mixed-Model Multiclass AIGT Detection:** This is a more difficult task, aiming not only to detect AIGT but also to identify the exact AI model behind it. Sniffer (Li et al., 2023) is a prominent model in this category.

## 2.3 Related Fields of Study

It's worth noting that AIGT detection has parallels with other security-centric studies such as watermarking (Kurita et al., 2020; Li et al., 2021; Kirchenbauer et al., 2023), harmful AI (Bai et al., 2022), and adversarial attacks (Jin et al., 2019; Li et al., 2020). These studies, together with AIGT detection, are indispensable for utilizing and safeguarding these advanced technologies and becoming more important in the era of LLMs.

## 3 Sentence-Level AIGT Detection

### 3.1 Sentence-Level Detection

The document-level detection methods struggle to accurately evaluate documents containing a mix of AI-generated and human-authored content or consisting of few sentences, which can potentially lead to a higher rate of false negatives or false positives.

In response to this limitation, we aim to study **sentence-level AIGT detection**. Sentence-level AIGT detection offers a fine-grained text analysis compared to document-level AIGT detection, as it explores each sentence within the entire text. By addressing this challenge, we could significantly reduce the risk of misidentification and achieve both higher detection accuracy and finer detection granularity than document-level detection.

Similarly, we study sentence-level detection challenges as follows:

1. Particular-Model Binary AIGT Detection: Discriminate whether each sentence within a candidate document is generated by a specific model or by a human.

2. Mixed-Model Binary AIGT Detection: Discriminate whether each sentence is generated by an AI model or a human, regardless of which AI model it is.

3. Mixed-Model Multiclass AIGT Detection: Discriminate whether each sentence is generated by an AI model or a human and identify which AI model generates it.

### 3.2 Dataset Construction

Current AIGT detection datasets are mainly designed for document-level detection. Therefore, we synthesize a sentence-level detection dataset for the study of fine-grained AIGT detection.

Since the human-authored documents from SnifferBench cover various domains including news articles from XSum (Narayan et al., 2018), social media posts from IMDB (Maas et al., 2011), web texts (Radford et al., 2019), scientific articles from PubMed and Arxiv(Cohan et al., 2018), and technical documentation from SQuAD (Rajpurkar et al., 2016), we utilize these documents to construct content that include both human-authored and AI-generated sentences. To ensure diversity while preserving coherence, we randomly select the first to third sentences from the human-generated documents as the prompt for subsequent AI sentence generation.

Specifically, when generating the AI component of a document using language models such as GPT-2, we utilize the prompt obtained via the above process to get the corresponding AI-generated sentences. In the case of instruction-tuned models like GPT-3.5-turbo, we provide specific instruction to assure the generation. We show instruction details in Appendix D.

Regarding open-source models, we gather the AI-generated portions of a document from GPT-2 (Radford et al., 2019), GPT-J (Wang and Komatsuzaki, 2021), GPT-Neo (Black et al., 2022), and LLaMA (Touvron et al., 2023). For models which provide only partial access, we collect data from GPT-3.5-turbo, an instruction-tuned model. With a collection of 6,000 human-written documents from SnifferBench, we similarly assemble 6,000 documents containing both human-written and AI-generated sentences from each of the models. This results in a total of 30,000 documents. Accordingly, we have:

**Particular-Model Binary Detection Dataset** To contrast with DetectGPT and the log $p(x)$ method, we construct the dataset using documents that consist of sentences generated by a particular model or by humans. For each document, sentences generated by the particular model are labeled as "AI", while sentences created by humans are labeled as "human". Using the collected data, we can construct five binary detection datasets in total.

**Mixed-Model Binary Detection Dataset** We utilize all the collected data to construct a dataset

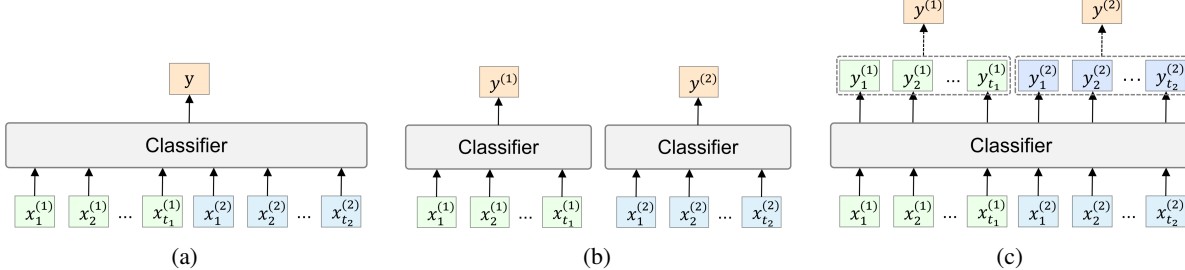

Figure 2: Strategies for AIGT Detection. (a) Document-Level AIGT Detection: Determine the category based on the entire candidate document. (b) Sentence Classification for Sentence-Level AIGT Detection: Classify each sentence one by one using the same model. (c) Sequence Labeling for Sentence-Level AIGT Detection: Classify the label for each word, then select the most frequently occurring category as the final category for each sentence.

intended for multi-model binary detection research. For each document, we label sentences generated by any model as "AI", and sentences created by humans as "human".

**Mixed-Model Multiclass Detection Dataset** For multiclass detection research, we again use the entire data. Unlike the binary detection dataset that merely differentiates between human-created and AI-generated sentences, we use more specific labels such as "GPT2", "GPTNeo", and so forth, instead of the generic label "AI". This allows us to further distinguish among a variety of AI models.

For each dataset, We divide the documents into a train/test split with a 90%/10% partition. We name the collected datasets **SeqXGPT-Bench**, which can be further used for fine-grained AIGT detection.

## 4 SeqXGPT

To solve the sentence-level detection challenge, we introduce a strong method **SeqXGPT**:

### 4.1 Strategies for Sentence-Level Detection

A sentence-level detection task is a specific form of the sentence classification task. Therefore, there are mainly two approaches:

**Sentence Classification** This method treats each sentence in the document as an input text and independently determines whether each sentence is generated by AI, as depicted in Figure 2(b).

**Sequence Labeling** This method treats the entire document as input, creating a sequence labeling task to decide the label for each word. For each sentence, we count the number of times each label appears and select the most frequent label as the category of the sentence, as shown in Figure 2(c). It's worth noting that each dataset uses a cross-label set based on $\{B, I, O\}$, similar to the usage in

sequence labeling tasks (Huang et al., 2015; Wang and Ren, 2022). We use $B$-AI, $B$-HUMAN, etc. as word-wise labels.

### 4.2 Baseline Approaches

In the sentence-level AIGT detection challenge, we explore how to modify baselines used in document-level detection:

**log $p(x)$:** We consider sentence-level detection as a sentence classification task to implement the log $p(x)$ method. The log $p(x)$ method discriminates AIGT through perplexity. Specifically, we utilize a particular model such as GPT-2 to compute the perplexity of each sentence in the document and draw a histogram (Figure 4) to show the distribution of perplexity. Then we select a threshold manually as the discrimination boundary to determine whether each sentence is generated by a human or an AI.

**DetectGPT** (Mitchell et al., 2023): Similarly, we regard sentence-level detection as a sentence classification task to implement DetectGPT. DetectGPT discriminates AIGT using their proposed z-score. For each sentence, we add multiple perturbations (we try 40 perturbations for each sentence). We then compute the z-scores for each sentence based on a particular model and generate a histogram (Figure 4) showing the score distribution. Similarly, we manually select a threshold to discriminate AI-generated sentences.

**Sniffer** (Li et al., 2023): Sniffer is a powerful model capable of detecting and tracing the origins of AI-generated texts. In order to conduct sentence-level AIGT detection, we train a sentence-level Sniffer following the structure and training process of the original Sniffer, except that our input is a single sentence instead of an entire document.

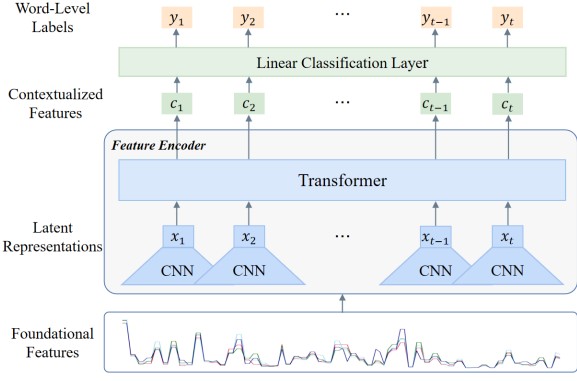

Figure 3: SeqXGPT Framework.

**RoBERTa**: RoBERTa (Liu et al., 2019) is based on the Transformer encoder that can perform sentence classification tasks as well as sequence labeling tasks. Based on RoBERTa-base, we train two RoBERTa baselines for sentence-level AIGT detection. As mentioned in Sec. 4.1, one model employs a sequence labeling approach, while the other uses a sentence classification method. They are referred to as Seq-RoBERTa and Sent-RoBERTa, respectively.

### 4.3 SeqXGPT Framework

In this section, we will provide a detailed introduction to SeqXGPT. The SeqXGPT employs a sequence labeling approach for sentence-level AIGT detection, which consists of the following three parts: (1) Perplexity Extraction and Alignment; (2) Feature Encoder; (3) Linear Classification Layer.

**Perplexity Extraction and Alignment**

Previous works demonstrate that both the average per-token log probability (DetectGPT) and contrastive features derived from token-wise probabilities (Sniffer) can contribute to AIGT detection. Therefore, in this work, we choose to extract token-wise log probability lists from several public open-source models, serving as the *original features* of SeqXGPT.

Specifically, given a candidate text $S$, a known model $\theta_n$ and the corresponding tokenizer $T_n$, we obtain the encoded tokens $\boldsymbol{x} = [x_1, ..., x_i, ...]$ and a list of token-wise log probabilities $ll_{\theta_n}(\boldsymbol{x})$, where the log probability of the $i^{th}$ token $x_i$ is the log-likelihood $ll_{\theta_n}(x_i) = \log p_{\theta_n}(x_i|x_{<i})$. Given a list of known models $\theta_1, ..., \theta_N$, we can obtain N token-wise log probability lists of the same text $S$.

Considering that the tokens are commonly sub-words of different granularities and that different tokenizers have different tokenization methods (e.g. byte-level BPE by GPT2 (Radford et al.,

2019), SentencePiece by LLaMA (Touvron et al., 2023)), they do not align perfectly. As a result, we align tokens to words, which addresses potential discrepancies in the tokenization process across different LLMs. We use a general word-wise tokenization of $S$: $\boldsymbol{w} = [w_1, \ldots, w_t]$ and align calculated log probabilities of tokens in $\boldsymbol{x}$ to the words in $\boldsymbol{w}$. Therefore, we can obtain a list of word-wise log probabilities $ll_{\theta_n}(\boldsymbol{w})$ from token-wise log probability list $ll_{\theta_n}(\boldsymbol{x})$. We concatenate these word-wise log probability lists together to form the *foundational features* $L = [l_1, \ldots, l_t]$, where the feature vector of the $i^{th}$ word $w_i$ is $l_i = [ll_{\theta_1}(w_i), \ldots, ll_{\theta_N}(w_i)]$.

We select GPT2, GPT-Neo, GPT-J, and LLaMA as white-box models to obtain the foundational features. In the various implementations of alignment algorithms, this paper utilizes the alignment method proposed by Li et al. (2023).

**Feature Encoder**

The foundational features are lists of word-wise log probabilities, which prevents us from utilizing any existing pre-trained models. Consequently, we need to design a new, efficient model structure to perform sequence labeling for AIGT detection based on these foundational features.

The word-wise log probability lists for the same sentence of different LLMs may differ due to different parameter scales, training data, and some other factors. Here, we propose treating the word-wise log probability list as a kind of feature that reflects a model's understanding of different semantic and syntactic structures. As an example, more complex models are able to learn more complex language patterns and syntactic structures, which can result in higher probability scores. Moreover, the actual log probability list might be subject to some uncertainty due to sampling randomness and other factors. Due to these properties, it is possible to view these temporal lists as **waves** in speech signals. Inspired by speech processing studies, we choose to employ convolutional networks to handle these foundational features. Using convolutional networks, local features can be extracted from input sequences and mapped into a hidden feature space. Subsequently, the output features are then fed into a context network based on self-attention layers, which can capture long-range dependencies to obtain *contextualized features*. In this way, our model can better understand and process the

wave-like foundational features.

As shown in Figure 3, the foundational features are first fed into a five-layer convolutional network $f : \mathcal{L} \rightarrow \mathcal{Z}$ to obtain the *latent representations* $[z_1, \ldots, z_t]$. Then, we apply a context network $g : \mathcal{Z} \rightarrow \mathcal{C}$ to the output of the convolutional network to build contextualized features $[c_1, \ldots, c_t]$ capturing information from the entire sequence. The convolutional network has kernel sizes $(5, 3, 3, 3, 3)$ and strides $(1, 1, 1, 1, 1)$, while the context network has two Transformer layers with the simple fixed positional embedding. More implementation details can be seen in Appendix A.

### Linear Classification Layer

After extracting the contextualized features, we train a simple linear classifier $F(\cdot)$ to project the features of each word to different labels. Finally, we count the number of times each word label appears and select the most frequent label as the final category of each sentence.

## 5 Experiments

### 5.1 Implementation Details

We conduct sentence-level AIGT detection based on SeqXGPT-Bench. As part of our experiments, we test modified existing methods as described in Sec. 4.2, as well as our proposed SeqXGPT detailed in Sec. 4.3.

We use four open-source (L)LMs to construct our SeqXGPT-Bench: GPT2-xl (1.5B), GPT-Neo (2.7B), GPT-J (6B) and LLaMA (7B). These models are also utilized to extract perplexity lists, which are used to construct the original features of our SeqXGPT and the contrastive features for Sniffer. For each model, we set up an inference server on NVIDIA4090 GPUs specifically for extracting perplexity lists. The maximum sequence length is set to 1024 for all models, which is the maximum context length supported by all the pre-trained models we used. We align all texts with a white-space tokenizer to obtain uniform tokenizations for our SeqXGPT.

In the evaluation process, we utilize Precision (P.), Recall (R.), and Macro-F1 Score as our metrics. Precision and Recall respectively reflect the "accuracy" and "coverage" of each category, while the Macro-F1 Score effectively combines these two indicators, allowing us to consider the overall performance.

### 5.2 Sentence-Level Detection Results

In this section, we present three groups of experiments:

**Particular-Model Binary AIGT Detection** We test zero-shot methods $\log p(x)$ and DetectGPT, as well as Sent-RoBERTa and our SeqXGPT on two datasets shown in Table 1. The results for the other two datasets can be found in the appendix in Table 6.

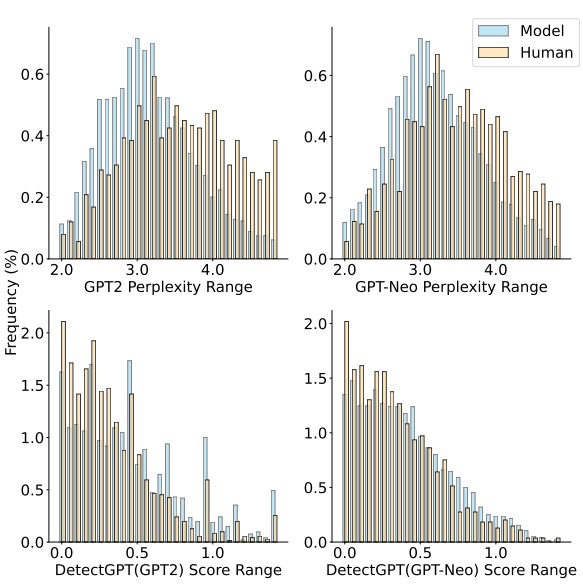

Figure 4: The discrepancy between AI-generated and human-created sentences. In each figure, different bars show different sentence origins and each figure is to use a certain model of a certain detect method to test given sentences. The top two figures use $\log p(x)$, while the bottom two use DetectGPT.

As seen in Figure 4, there is a large overlap between the peaks of AI-generated and human-created sentences when using $\log p(x)$ and DetectGPT, making it difficult to distinguish sentences from different categories. The peaks almost completely overlap for DetectGPT in particular. This situation can be attributed to two main factors. Firstly, the perplexity of a sentence tends to be more sensitive than a document. Secondly, given the brevity of sentences, perturbations often fail to achieve the same effect as in DetectGPT. That is, the perturbed sentences either remain nearly identical to the original, or vary significantly. These findings indicate that these two zero-shot methods are unsuitable for sentence-level AIGT detection.

Overall, SeqXGPT exhibits great performance on all datasets, including the two presented in

| Method | Different AIGT Origins | | | | | | | | | |
| | GPT-2 | | | | | GPT-Neo | | | | |
| | P.(AI) | R.(AI) | P.(H.) | R.(H.) | **Macro-F1** | P.(AI) | R.(AI) | P.(H.) | R.(H.) | **Macro-F1** |
|---|---|---|---|---|---|---|---|---|---|---|
| log $p(x)$ | 82.2 | 74.9 | 43.1 | 53.9 | 63.1 | 81.2 | 67.8 | 34.2 | 51.7 | 57.5 |
| DetectGPT | 80.9 | 55.4 | 32.7 | 62.4 | 54.3 | 82.6 | 44.2 | 29.1 | 71.2 | 49.4 |
| Sent-RoBERTa | 89.3 | 96.9 | 88.1 | 66.5 | 84.4 | 89.8 | 95.6 | 82.7 | 66.0 | 83.0 |
| SeqXGPT | **99.3** | **97.9** | **94.5** | **97.1** | **97.2** | **99.5** | **98.2** | **94.8** | **98.1** | **97.6** |

Table 1: Results of **Particular-Model Binary AIGT Detection** on two datasets with AI-generated sentences are from GPT-2 and GPT-Neo, respectively. The Macro-F1 is used to measure the overall performance, while the P. (precision) and R. (recall) are used to measure the performance in a specific category.

| Method | Different AIGT Origins | | | | | | | | | | | | |
| | GPT-2 | | GPT-2-Neo | | GPT-J | | LLaMA | | GPT-3 | | Human | | |
| | P. | R. | P. | R. | P. | R. | P. | R. | P. | R. | P. | R. | **Macro-F1** |
|---|---|---|---|---|---|---|---|---|---|---|---|---|---|
| Sniffer | 47.5 | 56.3 | 48.4 | 42.9 | 39.0 | 33.5 | 41.8 | 16.0 | 52.8 | 55.4 | 51.2 | 67.2 | 44.7 |
| Sent-RoBERTa | 38.6 | 48.9 | 36.9 | 27.6 | 34.9 | 28.7 | 57.5 | 33.6 | 65.5 | 97.1 | 89.4 | 91.6 | 52.9 |
| Seq-RoBERTa | 42.1 | 81.4 | 45.3 | 30.9 | 61.6 | 21.6 | 75.5 | 82.0 | 90.3 | **98.9** | **94.6** | 90.1 | 64.9 |
| SeqXGPT | **99.2** | **97.9** | **99.3** | **98.2** | **97.6** | **96.8** | **95.8** | **90.8** | **94.1** | 93.7 | 90.7 | **95.2** | **95.7** |
| w/o Transformer | 92.4 | 93.1 | 92.7 | 88.9 | 93.3 | 62.1 | 82.1 | 14.3 | 22.7 | 0.2 | 42.0 | 95.7 | 56.9 |
| w/o CNN | 0.0 | 0.0 | 0.0 | 0.0 | 0.0 | 0.0 | 0.0 | 0.0 | 0.0 | 0.0 | 24.8 | 100.0 | 6.6 |

Table 2: Results of **Mixed-Model Multiclass AIGC Detection.**

Table 1 and the other two in Table 6. Despite Sent-RoBERTa performs better than two zero-shot methods, it is still far inferior to our method.

**Mixed-Model Binary AIGT Detection** We implement Sniffer and Sent-RoBERTa based on sentence classification tasks, as well as our SeqXGPT and Seq-RoBERTa based on sequence labeling tasks. From the results in Table 3, it is apparent that our SeqXGPT shows the best performance among these four methods, significantly demonstrating the effectiveness of our method. In contrast, the performance of Sniffer is noticeably inferior, which emphasizes that document-level AIGT detection methods cannot be effectively modified for sentence-level AIGT detection. Interestingly, we find that the performance of both RoBERTa-based methods is slightly inferior to SeqXGPT in overall performance. This suggests that the semantic features of RoBERTa might be helpful to discriminate human-created sentences.

**Mixed-Model Multiclass AIGT Detection** As illustrated in Table 2, SeqXGPT can accurately discriminate sentences generated by various models and those authored by humans, demonstrating its strength in multi-class detection. It is noteworthy that RoBERTa-based methods perform significantly worse than binary AIGT detection. They are better at discriminating

| Method | Mixed AIGT Origins | | | | |
| | P.(AI) | R.(AI) | P.(H.) | R.(H.) | **Macro-F1** |
|---|---|---|---|---|---|
| Sniffer | 83.2 | 92.7 | 67.8 | 45.3 | 71.0 |
| Sent-RoBERTa | 97.3 | 97.9 | 93.5 | 91.8 | 95.1 |
| Seq-RoBERTa | 96.4 | **98.5** | **95.0** | 88.9 | 94.6 |
| SeqXGPT | **98.2** | 97.1 | 91.4 | **94.5** | **95.3** |

Table 3: Results of **Mixed-Model Binary AIGT Detection.**

sentences generated by humans or more human-like LLMs, such as GPT-3.5-turbo, while having difficulty detecting sentences generated by GPT-2/GPT-J/Neo. The reason for this may be that sentences generated by stronger models contain more semantic features. This is consistent with the findings of Li et al. (2023). In contrast, the Seq-RoBERTa outperforms Sent-RoBERTa, indicating that sequence labeling methods, with their ability to obtain the context information of the entire document, are more suitable for sentence-level AIGT detection. Sniffer, however, performs poorly, which may be due to its primary design focusing on document-level detection.

## 5.3 Document-Level Detection Results

We conduct experiments on a document-level detection dataset (more details in Appendix C.1). For sentence classification-based models, we classify each sentence and choose the category

| | Different AIGT Origins | | | | | | | | | | | |
| | GPT-2 | | GPT-2-Neo | | GPT-J | | LLaMA | | GPT-3 | | Human | | |
| Method | P. | R. | P. | R. | P. | R. | P. | R. | P. | R. | P. | R. | Macro-F1 |
|---|---|---|---|---|---|---|---|---|---|---|---|---|---|
| Sniffer | 76.5 | 96.6 | 86.0 | 83.1 | 75.0 | 74.2 | 92.9 | 7.0 | 79.5 | 83.1 | 53.4 | 87.2 | 67.5 |
| Sent-RoBERTa | 45.2 | 73.0 | 39.7 | 46.5 | 28.3 | 21.5 | 72.4 | 10.5 | 73.4 | **100.0** | 97.4 | 92.0 | 53.4 |
| Seq-RoBERTa | 50.4 | 85.5 | 42.1 | 40.0 | 42.4 | 26.5 | 62.6 | 72.0 | 85.7 | 99.0 | 85.9 | 36.5 | 57.9 |
| SeqXGPT | **100.0** | 99.0 | **100.0** | 99.0 | 99.5 | 96.5 | 96.8 | 90.0 | 94.5 | 86.5 | 77.6 | **93.5** | **94.2** |

Table 4: Results of **Document-Level AIGT Detection.**

| | Different AIGT Origins | | | | | | | | | | | |
| | GPT-2 | | GPT-2-Neo | | GPT-J | | LLaMA | | GPT-3 | | Human | | |
| Method | P. | R. | P. | R. | P. | R. | P. | R. | P. | R. | P. | R. | Macro-F1 |
|---|---|---|---|---|---|---|---|---|---|---|---|---|---|
| Sniffer | 4.1 | 80.0 | 59.9 | 44.1 | 21.5 | 41.2 | 54.9 | 14.5 | 73.0 | 53.7 | 35.8 | 60.0 | 36.1 |
| Sent-RoBERTa | 30.3 | 35.0 | 13.6 | 27.4 | 24.3 | 25.3 | 35.1 | 27.5 | 61.7 | 94.4 | 75.1 | 19.1 | 35.2 |
| Seq-RoBERTa | 46.2 | 64.2 | 22.7 | 40.1 | 60.7 | 19.8 | 74.5 | 75.9 | 86.2 | **99.3** | **89.5** | 78.4 | 60.6 |
| SeqXGPT | **99.5** | **98.4** | **99.1** | **83.6** | **95.5** | **95.0** | **91.6** | **89.1** | **96.5** | 91.0 | 83.1 | **94.0** | **92.8** |

Table 5: Results of **Out-of-Distribution Sentence-Level AIGT Detection.**

with the highest frequency as the document's category. For sequence labeling-based models, we label each word and select the category with the greatest number of appearances as the document's category.

Table 4 indicates that sentence-level detection methods can be transformed and directly applied to document-level detection, and the performance is positively correlated with their performance on sentence-level detection. Despite their great performance in detecting human-generated sentences, both Seq-RoBERTa and SeqXGPT perform slightly worse in discriminating human-created documents. This is mainly because our training set for human-created data only comes from the first 1-3 sentences, leading to a shortage of learning samples. Therefore, appropriately increasing human-generated data can further enhance the performance. We can see that Sniffer performs much better in document-level detection, indicating that Sniffer is more suitable for this task. Overall, SeqXGPT exhibits excellent performance in document-level detection.

### 5.4 Out-of-Distribution Results

Previous methods often show limitations on *out-of-distribution* (OOD) datasets (Mitchell et al., 2023). Therefore, we intend to test the performance of each method on the OOD sentence-level detection dataset (more details in Appendix C.2).

Table 5 demonstrates the great performance of SeqXGPT on OOD data, reflecting its strong generalization capabilities. Conversely, Sent-RoBERTa shows a significant decline in performance, especially when discriminating sentences generated by GPT-3.5-turbo and humans. This observation aligns with the findings from prior studies (Mitchell et al., 2023; Li et al., 2023), suggesting that semantically-based methods are prone to overfitting on the training dataset. However, Seq-RoBERTa performs relatively better on OOD data. We believe that this is because sequence labeling methods can capture contextualized information, helping to mitigate overfitting during learning. Despite this, its performance still falls short of SeqXGPT.

### 5.5 Ablations and Analysis

In this section, we perform ablation experiments on the structure of SeqXGPT to verify the effectiveness of different structures in the model.

First, we train a model using only transformer layers. As shown in Table 2, this model struggles with AIGT detection, incorrectly classifying all sentences as human-written. Although there are differences between a set of input feature waves (foundational features), they are highly correlated, often having the same rise and fall trend (Figure 3). Further, these features only have 4 dimensions. We think it's the high correlation and sparsity of the foundational features make it difficult for the Transformer layers to effectively learn AIGT detection. In contrast, CNNs are very good at handling temporal features. When combined

with CNNs, it's easy to extract more effective, concentrated feature vectors from these feature waves, thereby providing higher-quality features for subsequent Transformer layers. Moreover, the parameter-sharing mechanism of CNNs can reduce the parameters, helping us to learn effective features from limited samples. Thus, CNNs play an important role in SeqXGPT.

However, when only using CNNs (without Transformer layers), the model performs poorly on sentences generated by strong models such as GPT-3.5-turbo. The reason for this may be that CNNs primarily extract local features. In dealing with sentences generated by these strong models, it is not sufficient to rely solely on local features, and we must also consider the contextualized features provided by the Transformer layers.

## 6 Conclusion

In this paper, we first introduce the challenge of sentence-level AIGT detection and propose three tasks based on existing research in AIGT detection. Further, we introduce a strong approach, SeqXGPT, as well as a benchmark to solve this challenge. Through extensive experiments, our proposed SeqXGPT can obtain promising results in both sentence and document-level AIGT detection. On the OOD testset, SeqXGPT also exhibits strong generalization. We hope that SeqXGPT will inspire future research in AIGT detection, and may also provide insightful references for the detection of content generated by models in other fields.

## Limitations

Despite SeqXGPT exhibits excellent performance in both sentence and document-level AIGT detection challenge and exhibits strong generalization, it still present certain limitations:

(1) We did not incorporate semantic features, which could potentially assist our model further in the sentence recognition process, particularly in cases involving human-like sentence generation. We leave this exploration for future work.

(2) During the construction of GPT-3.5-turbo data, we did not extensively investigate the impact of more diversified instructions. Future research will dive into exploring the influence of instructions on AIGT detection.

(3) Due to limitations imposed by the model's context length and generation patterns, our samples only consist of two distinct sources of sentences.

In future studies, we aim to explore more complex scenarios where a document contains sentences from multiple sources.

## Acknowledgements

We would like to extend our gratitude to the anonymous reviewers for their valuable comments. Additionally, we sincerely thank Qipeng Guo for his valuable discussions and insightful suggestions on this study. This work was supported by the National Natural Science Foundation of China (No. 62236004 and No. 62022027).

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

| Method | Different AIGT Origins | | | | | | | | | |
| | GPT-J | | | | | LLaMA | | | | |
| | P.(AI) | R.(AI) | P.(H.) | R.(H.) | **Macro-F1** | P.(AI) | R.(AI) | P.(H.) | R.(H.) | **Macro-F1** |
|---|---|---|---|---|---|---|---|---|---|---|
| $\log p(x)$ | 80.2 | 73.2 | 30.5 | 39.4 | 55.5 | 70.2 | 68.1 | 26.1 | 28.1 | 48.1 |
| DetectGPT | 81.0 | 56.8 | 27.7 | 55.5 | 51.9 | 84.1 | 38.5 | 33.7 | 81.2 | 50.2 |
| Sent-RoBERTa | 89.7 | 96.7 | 84.7 | 62.3 | 82.4 | 86.8 | 92.7 | 77.0 | 63.5 | 79.6 |
| SeqXGPT | **98.0** | **97.8** | **92.6** | **93.3** | **95.4** | **97.3** | **94.7** | **87.0** | **93.1** | **92.9** |

Table 6: Results of **Particular-Model Binary AIGT Detection** on two datasets with AI-generated sentences are from GPT-J and LLaMA, respectively.

## A  Implementation Details

In this section, we will discuss the implementation details of the Feature Encoder, which is composed of a convolution network and a context network.

We use a five-layer convolution network, where the sizes of the convolution kernels are (5,3,3,3,3), the strides are (1,1,1,1,1), and the number of output channels are (64,128,128,128,64). Since SeqXGPT is trained using sequence labelling, the input and output must have the same length. We specify *padding='same'* in *torch.nn.Conv1d* to ensure that the lengths of the foundational features and latent representations are identical, despite the fact that the dimensions of each feature vector may differ.

Our Context Network consists of two transformer layers with 16 attention heads and a hidden size of 512. In implementation, we only use the simplest absolute position encoding as Vaswani et al. (2017).

## B  More Results

We provided results of *Particular-Model Binary AIGT Detection* on the other two datasets in Table 6. In fact, we have five binary detection datasets in total. Since Sniffer and DetectGPT can only perform white-box detection, we did not conduct experiment on dataset with AI-generated sentences are from GPT-3.5-turbo.

## C  Construction of Supplementary Datasets

In addition to the SeqXGPT-Bench designed specifically for sentence-level AIGT detection, we have also developed two additional datasets. These two datasets are respectively constructed for document-level AIGT detection and Out-of-Distribution (OOD) sentence-level AIGT detection. In the following, we will elaborate on the construction process of these two datasets.

### C.1  Document-Level Detection Dataset

In order to evaluate the performance in document-level AIGT detection, we construct a *document-level detection dataset*. To this end, we randomly sampled 200 human-created documents from the test set of SeqXGPT-Bench, which are used to construct AI-generated documents. Similar to Mitchell et al. (2023) and Li et al. (2023), we use the first sentence as a prompt to generate the entire document. This process is essentially the same as the construction process described in Sec. 3.2. The significant **difference** is that we set the maximum sequence length as large as possible and remove the prompt part after generation. We mix documents created by humans with those generated by AI to form our document-level detection dataset.

### C.2  OOD Sentence-Level Detection Dataset

We construct an *OOD sentence-level detection dataset* to evaluate the performance of various methods on OOD data. Our aim in creating this dataset is to ensure both topic-wise and format-wise diversity to best represent the possible challenges faced by AI-generated text detectors. With this goal in mind, we utilize the triviaQA dataset (Joshi et al., 2017) as the foundational data for our OOD dataset. This choice was made after careful consideration and deliberation.

TriviaQA covers a vast array of topics, drawing from a diverse collection of sources including blog articles, news reports, encyclopedia entries, and Wikipedia pages. This diversity ensures a comprehensive coverage of topics and introduces a rich variety of text styles and formats. Moreover, a significant advantage of utilizing triviaQA is its publication timeline. Published prior to the widespread emergence of AI text-generation technologies, the dataset is less likely to have been influenced by AI-generated texts, ensuring the authenticity of its human-derived content.

For the OOD dataset construction, we randomly

sample 200 evidence documents from triviaQA. These documents, with their wide-ranging domains and inherent diversity, form an excellent basis for testing OOD performance. The methodology adopted for constructing this dataset mirrors that detailed in Sec. 3.2, where we randomly select the first to third sentences from the human-generated documents as the prompt for subsequent AI generation. We then employ this newly constructed dataset as a testbed for evaluating the OOD performance of the models trained in Sec. 5.2.

## D   Instruction Details

The specific instruction used in GPT-3.5-turbo is:

*Please provide a continuation for the following content to make it coherent: {prompt}*

We consider simple instruction since we focus on studying the sentence-level AIGT detection without further analyzing the working mechanisms and better usage of instructions, which we leave to future works.