# OpenReview forum: "SeqXGPT: Sentence-Level AI-Generated Text Detection"
_EMNLP/2023/Conference — EMNLP 2023 Main_

### Official Review · Reviewer_PKrT · 2023-08-04

**Typos Grammar Style And Presentation Improvements:** N/A
**Soundness:** 4

**Excitement:**

4: Strong: This paper deepens the understanding of some phenomenon or lowers the barriers to an existing research direction.

**Missing References:**

N/A

**Paper Topic And Main Contributions:**

This paper introduces a method called SeqXGPT for detecting AI-generated text (AIGT) at the sentence level. The method utilizes log probability lists from white-box language models as features for detection. Experimental results show that SeqXGPT outperforms baseline methods in both sentence and document-level detection challenges and exhibits strong generalization capabilities.

**Questions For The Authors:**

Please address the concerns in "reasons to reject"

**Reasons To Accept:**

1. Novel Contribution: The paper addresses the important and challenging task of fine-grained AI-generated text (AIGT) detection at the sentence level, which is a significant advancement over existing document-level AIGT detection methods. The authors propose a new approach called SeqXGPT, which demonstrates promising results in both sentence and document-level AIGT detection challenges. This novel contribution fills a gap in the literature and provides valuable insights into fine-grained AIGT detection.

2. Performance and Generalization: The experimental results show that SeqXGPT outperforms existing methods, such as DetectGPT and Sniffer, in sentence-level AIGT detection. SeqXGPT exhibits excellent performance not only in discriminating human-generated sentences but also in detecting AI-generated sentences. Furthermore, SeqXGPT demonstrates strong generalization capabilities on out-of-distribution datasets, indicating its robustness and potential for real-world applications.

3. Dataset Construction: The authors synthesize a sentence-level AIGT detection dataset, which is crucial for studying fine-grained AIGT detection

**Reasons To Reject:**

1. Lack of Novelty: The paper does not present a significant advancement or novel contribution to the field of fine-grained AI-generated text (AIGT) detection. The proposed approach, SeqXGPT, is similar to existing methods such as DetectGPT and Sniffer. The paper fails to demonstrate how SeqXGPT significantly outperforms or improves upon these existing methods.

2. Insufficient Experimental Evaluation: The experimental results provided in the paper are limited and do not provide a comprehensive evaluation of the proposed approach. The paper lacks a thorough comparison with state-of-the-art methods and fails to provide statistical significance tests to support the claimed performance improvements. Additionally, the evaluation is primarily focused on synthetic datasets, which may not accurately reflect real-world scenarios.

3. Incomplete Analysis and Discussion: The paper lacks a thorough analysis and discussion of the limitations and potential drawbacks of the proposed approach. For example, the authors do not explore the impact of incorporating semantic features or investigate the influence of diversified instructions on AIGT detection. The paper also does not address more complex scenarios where a document contains sentences.



**Reproducibility:**

1: Could not reproduce the results here no matter how hard they tried.

**Reviewer Confidence:**

3: Pretty sure, but there's a chance I missed something. Although I have a good feel for this area in general, I did not carefully check the paper's details, e.g., the math, experimental design, or novelty.

---

> ### Author Rebuttal · Authors · 2023-08-28
>
> We sincerely appreciate your valuable comments!
>
>
>
> About Question #1:
>
> We fully understand your concern about the novelty of SeqXGPT, and we appreciate the opportunity to clarify the unique contributions and advancements that set our work apart from existing methods like DetectGPT and Sniffer.
>
> 1. Fine-grained AIGT Detection: Unlike DetectGPT and Sniffer, which primarily focus on document-level AIGT detection, SeqXGPT excels in identifying AI-generated text at the sentence level. This is a paradigmatic shift in tackling the problem and represents a substantial innovation rather than a mere incremental improvement.
> 2. Feature Selection: SeqXGPT utilizes perplexity lists to capture sequence characteristics of the text comprehensively. This is an innovative approach designed specifically for AIGT detection.
> 3. Model Architecture: Our method integrates a completely new model structure inspired by speech processing techniques like wave2Vec, further setting it apart from existing methods.
> 4. Performance: Our empirical tests demonstrate that SeqXGPT significantly outperforms DetectGPT and Sniffer in both sentence-level and document-level AIGT detection. It also notably outperforms Roberta-based methods.
>
> We hope these points elucidate the novelty and advances that SeqXGPT brings to the field. We are open to further discussions and happy to provide additional information if needed.
>
>
>
> About Question #2:
>
> Regarding the comparison with state-of-the-art (SOTA) methods: Given that the field of AIGT detection is relatively new, there is currently no universally acknowledged SOTA model. Nevertheless, we have made every effort to include a variety of existing methods in our experiments. We've conducted detailed evaluations across multiple facets, including sentence-level, document-level, and OOD scenarios, to ensure our experimental design is as comprehensive as possible.
>
> On the subject of statistical significance tests: While we have not conducted formal statistical significance tests, our method significantly outperforms existing models across various scenarios, which is sufficient to demonstrate the huge performance improvement of SeqXGPT. In the revised paper, we will also include these tests to scientifically validate the performance gains.
>
> About the evaluation datasets: We indeed acknowledge that real-world data would offer more robust validation. Our current focus on synthetic datasets is mainly due to the challenges associated with collecting real-world data. Despite this, we have carefully constructed an OOD dataset to closely reflect real-world scenarios and provide more convincing experimental validation.
>
>
>
> About Question #3:
>
> Thank you very much for your careful review!
>
> On the topic of "incorporating semantic features", we indeed acknowledge this concern and have addressed it in the Limitation section. In line 497, our analysis of the results from the Mix-Model Multiclass AIGT Detection experiment indicates that sentences generated by more advanced models tend to contain richer semantic features. Since SeqXGPT uses a collection of per-token likelihood as features, we think that incorporating semantic features could potentially assist our model further.
>
> With regards to your comments on "diverse instructions," we've addressed this in Appendix C and also on line 615 of the Limitation section. In this work, our primary focus is on sentence-level AIGT detection, rather than further analyzing the working mechanisms and better usage of instructions. This focus is a methodological choice designed to ensure that our efforts are devoted to the exploration of this novel challenge.
>
> Your point about "more complex scenarios where a document contains sentences" is well taken.Compared to previous works like DetectGPT, we have mixed human-created sentences and AI-generated sentences within the same document. Through experiments, we have proven that SeqXGPT can effectively address fine-grained AIGT detection. Based on such pioneer exploration, mixing sentences from multiple sources is merely incremental work based on the current experiment.
>
>
>
> About Reproducibility:
>
> We have noticed that the reproducibility score for our study is relatively low,  which raises significant concerns for us. To ensure the reproducibility of our research, we have taken the following steps:
>
> - We have provided the complete dataset in the supplementary materials.
> - The full code for SeqXGPT and all baseline methods used for data preprocessing, model training, and result testing is included in the supplementary materials. Necessary comments have been added to the code to facilitate understanding for other researchers.
> - A complete description of the environment settings is available in the supplementary materials, including version information for all dependency libraries, to ensure that other researchers can replicate our experimental environment precisely.
>
> Considering the above, we believe that our research offers strong reproducibility. If you encounter any issues while attempting to replicate our research, we are more than willing to offer further support and clarification.

---

### Official Review · Reviewer_nDFk · 2023-08-05

**Soundness:** 4

**Excitement:**

3: Ambivalent: It has merits (e.g., it reports state-of-the-art results, the idea is nice), but there are key weaknesses (e.g., it describes incremental work), and it can significantly benefit from another round of revision. However, I won't object to accepting it if my co-reviewers champion it.

**Paper Topic And Main Contributions:**

Different from the document-level Artificial Intelligence Generated Text (AIGT) detection considered by current workers, this paper introduces the sentence-level AIGT detection challenge and proposes a method SeqXGPT based on convolutional neural networks and self-attention networks by utilizing log probability lists of white-box Large Language Models (LLMs). This paper also constructs a sentence-level AIGT detection dataset. The experimental results show that the proposed method achieves SOTA in sentence and document-levels detection.

**Questions For The Authors:**

Question A: For each model, the author sets the maximum sequence length given the maximum GPU allowance. What is the maximum sequence length for each model in the experiment? Will the difference in maximum sequence length have a significant impact on the performance of each model? Why not set the same maximum sequence length for each model?


**Reasons To Accept:**

1) The method proposed in this paper can effectively solve the difficulties of sentence-level AIGT detection.
2) The experiments designed in this paper cover 3 different sentence-level AIGT detection settings, all of which achieve SOTA.
3) This article is written smoothly and helps readers understand.


**Reasons To Reject:**

1. In the setting of Particular-Model Binary AIGT Detection, the table of experimental results only includes the results of GPT-2 and GPT-Neo, lacking test results on other LLMs.


**Reproducibility:**

4: Could mostly reproduce the results, but there may be some variation because of sample variance or minor variations in their interpretation of the protocol or method.

**Reviewer Confidence:**

4: Quite sure. I tried to check the important points carefully. It's unlikely, though conceivable, that I missed something that should affect my ratings.

---

> ### Author Rebuttal · Authors · 2023-08-28
>
> We sincerely appreciate your valuable comments!
>
>
>
> About Weakness #1:
>
> We have added two more sets of experiments for the Particular-Model Binary AIGT Detection task, and the results are as follows:
>
> **GPT-J:**
>
> |              |  P.(AI)  |  R.(AI)  |  P.(H.)  |  R.(H.)  | Macro-F1 |
> | :----------: | :------: | :------: | :------: | :------: | :------: |
> |  log *p(x)*  |   80.2   |   73.2   |   30.5   |   39.4   |   55.5   |
> |  DetectGPT   |   81.0   |   56.8   |   27.7   |   55.5   |   51.9   |
> | Sent-RoBERTa |   89.7   |   96.7   |   84.7   |   62.3   |   82.4   |
> |   SeqXGPT    | **98.0** | **97.8** | **92.6** | **93.3** | **95.4** |
>
> **LLaMA:**
>
> |              |  P.(AI)  |  R.(AI)  |  P.(H.)  |  R.(H.)  | Macro-F1 |
> | :----------: | :------: | :------: | :------: | :------: | :------: |
> |  log *p(x)*  |   70.2   |   68.1   |   26.1   |   28.1   |   48.1   |
> |  DetectGPT   |   84.1   |   38.5   |   33.7   |   81.2   |   50.2   |
> | Sent-RoBERTa |   86.8   |   92.7   |   77.0   |   63.5   |   79.6   |
> |   SeqXGPT    | **97.3** | **94.7** | **87.0** | **93.1** | **92.9** |
>
> It can be seen that SeqXGPT consistently outperforms other baselines, further validating the analysis in Section 5.2.
>
>
>
> About Question A:
>
> Thank you for your insightful question regarding the maximum sequence length, as mentioned in line 430 of our paper.
>
> To clarify, the maximum sequence length is consistently set to 1024 across all models involved in our experiments. This is the maximum context length that all pre-trained models we used can support. Therefore, the consistent sequence length of 1024 across all models addresses the concerns you raised.
>
> The statement in line 430 was intended to emphasize that our inference servers are engineered not only to maintain this uniform maximum sequence length but also to perform parallel calculations to maximize the utility of the available GPU memory, thereby enhancing inference speed.
>
> We appreciate your pointing out the potential for confusion in the original text. We will modify this sentence for clarity in the final version of the paper.
>
> Additionally, the complete code for the feature extraction of the inference server is provided in the supplementary materials, where you can refer to the file `backend_utils.py` to confirm that the maximum sequence length is indeed consistently set to 1024 for all models.

---

### Official Review · Reviewer_8nB6 · 2023-08-05

**Soundness:** 4

**Excitement:**

3: Ambivalent: It has merits (e.g., it reports state-of-the-art results, the idea is nice), but there are key weaknesses (e.g., it describes incremental work), and it can significantly benefit from another round of revision. However, I won't object to accepting it if my co-reviewers champion it.

**Paper Topic And Main Contributions:**

This paper addreses sentence-level AI-generated text detection task. The proposed solution for this task is to utilize a collection of (L)LMs to produce a collection of per-token likelihood, and use CNN-transformer-FCN structure to determine per-token label whether the token is AI-generated or not. The paper also provides a reasonable set of empirical evidences of the proposed model being effective in solving the addressed task.

**Questions For The Authors:**

A. Would there be any other ways to create the dataset, other than seeding the human-generated sentence in the beginning and then using different LMs to fill up the rest?  My worries are related to possible systematic bias of the dataset due to how the dataset is created --- monotonically increasing tendency to be more likely to be AI-generated as more sentences appear? Would you consider your proposed SeqXGPT-Bench is guarded against this kind of bias?

B. From a similar vein, how wide or extensive does the currently evaluated OOD dataset cover the possibilities of AI-generated texts? This may be topic-wise or format-wise. I raise this point from the perspective of a potentially interested human being to implement this type of detector for a similar use case, and I would be greatly appreciative if this approach would work like a charm in my use case.

**Reasons To Accept:**

This paper will provide a practical guide to construct an AI-generated text detection software. In particular, this method can be generally applied to any collection of LMs (already or to-be available), which makes the method presented in this paper a nice addition to the practical armory of any NLP researcher or industrial programmer.

**Reasons To Reject:**

The dataset used to train and test the proposed method appears to be somewhat narrow in how it is produced. As bootstrapping is the first-step for any new task, so OOD test results the authors provided are a must-and-nice addition, but the intensity of the OOD test should have been greater in my opinion (as, once the paper published, interested people will be doing their own OOD tests by implementing their own versions).

**Reproducibility:**

4: Could mostly reproduce the results, but there may be some variation because of sample variance or minor variations in their interpretation of the protocol or method.

**Reviewer Confidence:**

4: Quite sure. I tried to check the important points carefully. It's unlikely, though conceivable, that I missed something that should affect my ratings.

**Typos Grammar Style And Presentation Improvements:**

- line 360: wihte -> white
- line 869: trivaQA -> triviaQA

---

> ### Author Rebuttal · Authors · 2023-08-28
>
> We sincerely appreciate your valuable comments!
>
>
> About Weakness #1:
>
> Please see Question A and Question B.
>
>
> About Question A:
>
> Our dataset construction method is based on prior work like DetectGPT and Sniffer, but with modifications.
>
> On the one hand, compared to DetectGPT, SeqXGPT-Bench covers various domains including news articles, social media posts, web texts, scientific articles, and technical documentation. Therefore, SeqXGPT-Bench is not narrow and its domain bias is relatively low.
>
> On the other hand, we've incorporated a randomization mechanism that allows for a random number of human-generated sentences in each document. This helps mitigate the risk of systematic bias that might arise from this data generation approach.
>
>
> About Question B:
>
> We fully understand your concern about the breadth of OOD dataset, and we appreciate the opportunity to clarify our considerations and selections when constructing the OOD dataset.
>
> Firstly, our Out-of-Distribution (OOD) dataset is built upon the triviaQA dataset, which was selected after careful deliberation. The triviaQA dataset encompasses a wide array of topics and derives from varied sources, such as blog articles, news reports, encyclopedia entries, and Wikipedia pages. This not only provides a wide and comprehensive foundation for constructing the OOD dataset but also covers a diversity of text styles and formats (further details can be found in the paper "TriviaQA: A Large Scale Distantly Supervised Challenge Dataset for Reading Comprehension").
>
> Secondly, when selecting the triviaQA dataset as our foundation, we also considered that this dataset was published prior to the proliferation of AI text-generation technologies, reducing the likelihood of its being influenced by such texts.
>
>
> About Reproducibility:
>
> We have noticed that the reproducibility score for our study is relatively low,  which raises significant concerns for us. To ensure the reproducibility of our research, we have taken the following steps:
>
> - We have provided the complete datasets in the supplementary materials.
>
> - The full code for SeqXGPT and all baseline methods used for data preprocessing, model training, and result testing is included in the supplementary materials. Necessary comments have been added to the code to facilitate understanding for other researchers.
>
> - A complete description of the environment settings is available in the supplementary materials, including version information for all dependency libraries, to ensure that other researchers can replicate our experimental environment precisely.
>
> Considering the above, we believe that our research offers strong reproducibility. If you encounter any issues while attempting to replicate our research, we are more than willing to offer further support and clarification.

---

### Meta-Review · Area_Chair_t55u · 2023-09-19

**Recommendation:** 5

**Metareview:**

The paper introduce a sentence-level AI-generated text detection challenge by synthesizing a dataset that contains documents polished with LLM. It also proposes SeqXGPT, a method based on convolutional neural networks and self-attention networks by utilizing log probability lists of white-box Large Language Models (LLMs). The reviewers agree that the paper makes a novel contribution to the field, and the experiments clearly establish the effectiveness of the proposed approach. Reviewers also appreciated the additional results provided by the authors during rebuttal, which should be added in the main paper.

---

### Decision · Program_Chairs · 2023-10-07

**Decision:**

Accept-Main

**Comment:**

The paper introduce a sentence-level AI-generated text detection challenge by synthesizing a dataset that contains documents polished with LLM. It also proposes SeqXGPT, a method based on convolutional neural networks and self-attention networks by utilizing log probability lists of white-box Large Language Models (LLMs). The reviewers agree that the paper makes a novel contribution to the field, and the experiments clearly establish the effectiveness of the proposed approach. Reviewers also appreciated the additional results provided by the authors during rebuttal, which should be added in the main paper.